# Use of Epinephrine in Cardiac Arrest: Advances and Future Challenges

**DOI:** 10.3390/medicina60111904

**Published:** 2024-11-20

**Authors:** Caitlin A. Williams, Hannah E. Fairley, Quincy K. Tran, Ali Pourmand

**Affiliations:** 1Department of Emergency Medicine, George Washington University School of Medicine and Health Sciences, Washington, DC 20052, USA; cawilliams@gwu.edu (C.A.W.); hfairley@gwu.edu (H.E.F.); pourmand@gwu.edu (A.P.); 2Department of Emergency Medicine, University of Maryland School of Medicine, Baltimore, MD 21201, USA; 3Program in Trauma, The R Adams Cowley Shock Trauma Center, University of Maryland School of Medicine, Baltimore, MD 21201, USA

**Keywords:** cardiac arrest, cardiopulmonary resuscitation, epinephrine, return of spontaneous circulation

## Abstract

Epinephrine is the most common medication used in cardiac arrest. Although the medication has been a mainstay of treatment over the last century, the utility and efficacy of epinephrine has been re-evaluated in recent years. This study aims to evaluate the literature describing the efficacy, timing, and dosing of epinephrine use in cardiac arrest. We utilized an extensive PubMed and SCOPUS search that included randomized control trials, prospective observational studies, and secondary analysis of observational data. These articles evaluated the administration of epinephrine in cardiac arrest and reported patient outcomes, including survival rates, neurological function, and return of spontaneous circulation. Dosing of epinephrine has been standardized at 1 mg per administration in adults and studies show that higher doses may not have better outcomes and can potentially be harmful. Research on the optimal timing of epinephrine has shown that earlier administration of epinephrine in cardiac arrest is more likely to have improved outcomes compared to later administration and longer intervals, although there are still conflicting results on the improvement of neurological outcomes. Intravenous is the preferred route of administration for epinephrine, but new research suggests intramuscular administration may be beneficial. While epinephrine has been shown to improve the rates of return of spontaneous circulation and even survival to hospital discharge in several studies, epinephrine use may not provide patients who survive cardiac arrest with a meaningful neurological recovery.

## 1. Introduction

Cardiac arrest is a leading cause of death worldwide. In the United States, survival rates to hospital discharge after out-of-hospital cardiac arrest (OHCA) are approximately 9% [1]. Furthermore, since morbidity is high in cardiac arrest, it places a significant financial and labor burden on the healthcare system, particularly due to prolonged hospital stays and complex post-arrest care [2]. Treatment for cardiac arrest includes early initiation of cardiopulmonary resuscitation (CPR), that includes high quality chest compressions, prompt defibrillation in the case of shockable cardiac arrhythmias like ventricular fibrillation and pulseless ventricular tachycardia, as well as pharmacologic intervention [3].

Epinephrine is a key medication in resuscitation protocols in patients experiencing cardiac arrest, both OHCA and in-hospital cardiac arrest (IHCA), in the United States. Exogenous epinephrine performs as a catecholamine that acts on both alpha and beta-adrenergic receptors [4]. Epinephrine leads to vasoconstriction through its effect on vascular smooth muscle alpha-1 receptors, while increased heart rate and myocardial contractility are a function of beta-1 receptors on cardiac myocytes [4]. Through these vasoconstrictive and inotropic properties, epinephrine use in cardiac arrest is thought to help improve both coronary and cerebral perfusion [4].

The adult cardiac arrest algorithm, also known as the Advanced Cardiac Life Support (ACLS) guidelines created by the American Heart Association (AHA), currently recommends the use of epinephrine every 3 to 5 min [5]. Standard dosing of epinephrine in resuscitation of adults experiencing cardiac arrest is 1 milligram (mg) administered either via intravenous (IV) access or intraosseous (IO) vascular access [5].

Despite the widespread use of epinephrine, its effectiveness, optimal dosing, and timing of administration have been controversial. Results of studies over the years have raised concerns regarding potential adverse outcomes of epinephrine use, including poor neurological recovery and survival to hospital discharge. This review seeks to explore the current evidence regarding the utility of epinephrine in cardiac arrest, while addressing the clinical effectiveness and optimal dosing and timing of its administration.

## 2. Methods

To identify relevant studies for this narrative review on therapeutic interventions involving the use of epinephrine in cardiac arrest, a comprehensive search of the PubMed database was conducted. The search spanned from the database’s inception to September 2024. The search strategy utilized a combination of Medical Subject Headings (MeSH) terms and filters to focus on pertinent literature. Specifically, the terms “((“Heart Arrest” [MeSH Terms] OR “Cardiac Arrest” [MeSH Terms]) AND “Epinephrine” [MeSH Terms]) AND ((english[Filter]) AND (alladult[Filter]))” were applied to narrow the results to studies involving adults and published in the English language.

### 2.1. Inclusion and Exclusion Criteria

Our review focused on studies that met stringent inclusion criteria including adult cardiac arrest patients who received epinephrine during their resuscitation. We included randomized controlled trials (RCTs), prospective observational studies, and secondary analyses of prospective observational data. We also included retrospective studies to strengthen our arguments regarding supplementary interventions in cardiac arrest management, such as extracorporeal cardiopulmonary resuscitation (ECPR). The primary criterion was that these studies evaluated therapeutic interventions involving the administration of epinephrine during cardiac arrest and reported patient-related outcomes. These outcomes could include survival rates, neurological function, return of spontaneous circulation (ROSC), and other relevant clinical endpoints that assessed the effectiveness and safety of epinephrine in cardiac arrest.

Studies that did not meet these inclusion criteria were excluded. Specifically, studies were excluded if they did not report patient-centered outcomes, such as those focused solely on biomarker levels or physiological responses without a connection to patient survival or clinical prognosis. Other excluded studies were non-original publications, such as reviews, meta-analyses, case reports, or conference abstracts that lacked sufficient detail or peer review. Studies not available in full-text English were also excluded, as this review was limited to studies published in English for consistency and accessibility. Studies prior to 1990 were also excluded. Furthermore, as this review focused on cardiac arrest in adult patients, studies with participants only under the age of 18 were excluded.

### 2.2. Screening and Selection Process

To ensure the rigor and accuracy of the review, two investigators independently screened the titles and abstracts of all studies retrieved from the PubMed search. Any discrepancies between the two investigators were adjudicated by a third investigator. Agreement from at least two of the three investigators was required for a study to be included in the review. This triage process ensured that all studies included in the final review met the predetermined inclusion criteria, minimized bias, and improved the reliability of the screening process. As this review was based on previously published research and did not involve the collection of new data or the enrollment of human subjects, the Institutional Review Board was not considered at our institution. All included studies had already undergone ethical review and approval, where applicable, at the primary authors’ respective institutions.

## 3. Results

### 3.1. Efficacy and Long-Term Outcomes (Table 1)

Cardiac arrest results in cessation of blood flow to all major organs, particularly the brain. Prolonged resuscitation and no flow state, the so called “down-time” period, can lead to cerebral hypoxia by primary ischemia with cellular damage and neuronal injury [6]. With ROSC and reperfusion of the brain, there is intracellular calcium release prompting the release of lytic enzymes and mitochondrial dysfunction, as well as activation of the immune system and resultant inflammation, which further leads to cellular damage [6].

As such, neurological outcomes in patients who achieve ROSC vary and depend on many factors, including duration of the arrest and the quality of resuscitation. Furthermore, post-cardiac arrest states are often complicated by multi-organ failure, which compromises patients’ recovery [7]. Recent research has aimed to determine if epinephrine use in cardiac arrest improves outcomes, including functional neurological status. Limited RCTs have evaluated the efficacy of epinephrine in cardiac arrest, including rates of ROSC, survival to hospital discharge, and ultimate neurological outcome.

An RCT in Australia in 2011 compared the use of epinephrine to placebo in OHCA in the pre-hospital setting and found that the likelihood of achieving ROSC in patients who received epinephrine was 3.4 times greater than for those who received the placebo [8]. However, there were no statistically significant differences in neurological outcome nor survival to hospital discharge [8]. Similarly, a nonrandomized prospective observational study conducted in Japan in 2012 also compared prehospital use of epinephrine to placebo in cardiac arrest and found that while epinephrine was associated with higher rates of ROSC before hospital arrival, there was a lower chance of survival and good functional outcomes at 1 month after the cardiac arrest [9]. When compared to no epinephrine use, outcomes were similar with greater rates of short-term survival, but decreased survival to hospital discharge and neurological outcome after OHCA [10]. A later cohort study also determined that epinephrine was negatively associated with a favorable neurological outcome, after adjusting for confounding variables [11].

A 2018 RCT in the UK compared the administration of 1 mg of epinephrine to normal saline as placebo during medication dosing increments of every 3 to 5 min, which was recommended per European Resuscitation Council Guidelines [12]. Patients that received epinephrine had higher rates of ROSC and admission to the ICU, although there was no statistically significant difference in neurological outcomes [12].

These results suggest that when comparing epinephrine to placebo, while there is the potential to improve rates of ROSC, there is at best no difference in neurological outcome and, in some cases, a potential for worse neurological outcomes in patients receiving epinephrine in cardiac arrest. Epinephrine administration has also been linked to higher rates of abnormal cardiac rhythm after ROSC, including ventricular tachycardia and ventricular fibrillation [13], which may further contribute to poor outcomes.

**Table 1 medicina-60-01904-t001:** Efficacy of epinephrine for different outcomes among patients with cardiac arrest.

Author	Year	Country	Setting	Initial Cardiac Rhythm	Design	Primary Outcome	Primary Findings	Additional Outcomes	Additional Findings	Conclusion
Jacobs, et al. [8]	2011	Australia	OHCA	Non-Shockable, Shockable	Randomized Control Trial	Survival to Hospital Discharge	4.0% epi vs. 1.9% (OR = 2.2; 95% CI, 0.7 to 6.3; *p*-value: 0.15)	ROSC, Survival to Hospital Admission, Neurological Status at Discharge	ROSC: 23.5% epi vs. 8.4% (OR = 3.4; 95% CI, 2.0 to 5.6; *p*-value < 0.001) Survival to Admission: 25.4% epi vs. 13.0% (OR = 2.3; 95% CI, 1.4 to 3.6; *p*-value < 0.001) Functional Neuro Status of Survivors: 81.8% epi vs. 100% (*p*-value = 0.31)	There was a statistically significant difference in rates of ROSC and survival to hospital admission in patients receiving epinephrine, but no statistically significant difference in survival to hospital discharge (primary outcome) and functional neurological status at discharge.
Hagihara, et al. [9]	2012	Japan	OHCA	Non-Shockable, Shockable	Prospective, nonrandomized, observational propensity analysis	ROSC before Hospital Admission	18.5% epi vs. 5.7% no-epi (OR = 2.36; 95% CI, 2.22 to 2.50; *p*-value < 0.001)	Survival at 1 Month, Cerebral Performance Category 1–2 (CPC), Overall Performance Category 1–2 (OPC)	Survival at 1 Month: 15.4% epi vs. 21.3% no-epi (OR = 0.46; 95% CI, 0.42 to 0.51; *p*-value < 0.001) CPC: 6.1% epi vs. 13.5% no-epi (OR = 0.31; 95% CI, 0.26 to 0.36; *p*-value < 0.001) OPC: 6.2% epi vs. 13.5 no-epi (OR = 0.32; 95% CI, 0.27 to 0.38; *p*-value < 0.001	While epinephrine was associated with higher rates of ROSC before hospital arrival, patient’s receiving epinephrine had a lower chance of survival and good functional outcomes 1 month after the cardiac arrest.
Olasveengen, et al. [10]	2012	Norway	OHCA	Non-Shockable, Shockable	Prospective Cohort Study	Survival to hospital discharge	23% epi vs. 56% no-epi (OR = 0.5; 95% CI, 0.3 to 0.8; *p* = 0.006)	Admission to Hospital, Favorable Neurological Outcome, Survival at One-Year after Cardiac Arrest,	Admission to hospital: 48% epi vs. 27% no-epi (OR = 2.5; 95% CI, 1.9 to 3.4; *p* < 0.001) Favorable Neuro Outcome: 5% epi vs. 11% no-epi (OR = 0.4; 95% CI, 0.2 to 0.7; *p* = 0.001) Survival at One-Year Post-Cardiac Arrest: 6% epi vs. 12% no-epi (OR = 0.5; 95% CI, 0.3 to 0.8; *p* = 0.004)	While patients receiving epinephrine were more likely to survive to hospital admission compared to those not receiving it, they were less likely to have a favorable neuro outcome or survive at one-year.
Dumas, et al. [11]	2014	France	OHCA	Non-Shockable, Shockable	Retrospective Cohort Study	Favorable Neurological Outcome at Discharge (CPC 1 or 2), Epinephrine vs. Placebo	17% epi vs. 60% no-epi (OR = 0.14; 95% CI, 0.10 to 0.17; *p*-value < 0.001)	Favorable Neurological Outcome at Discharge by Epinephrine Dosing	Neuro Outcome 1 mg Epi: 31% epi vs. 60% placebo (aOR = 0.48; 95% CI, 0.27 to 0.84; *p*-value = 0.01) Neuro Outcome 2–5 mg Epi: 18% epi vs. 60% placebo (aOR = 0.30; 95% CI, 0.20 to 0.47; *p*-value < 0.001) Neuro Outcome >5 mg Epi: 12% epi vs. 60% placebo (aOR = 0.23; 95% CI, 0.14 to 0.37; *p*-value < 0.001)	Epinephrine administration was negatively associated with a favorable neurological outcome, even after adjusting for confounding variables. This association also varied by dose of epinephrine.
Perkins, et al. [12]	2018	United Kingdom	OHCA	Non-Shockable, Shockable	Randomized Control Trial	Rate of Survival at 30 Days	3.2% epi vs. 2.4% placebo (OR = 1.39; 95% CI, 1.06 to 1.8; *p*-value = 0.02)	Survival until Hospital Admission, Survival until Hospital Discharge, Favorable Neurologic Outcome at Discharge	Survival until hospital admission: 23.8% epi vs. 8.0% placebo (OR = 3.59; 95% CI, 3.14 to 4.12) Survival until hospital discharge: 3.2% epi vs. 2.3% placebo (OR = 1.41; 95% CI, 1.0 to 1.86) Functional neuro outcome: 2.2% epi vs. 1.9% placebo (OR = 1.18; 95% CI, 0.86 to 1.61)	Administration of epinephrine compared to placebo had statistically higher rates of ROSC and survival at 30 days, but no statistically significant difference in functional neurological outcome.
Neset, et al. [13]	2013	Norway	OHCA	Non-shockable, Shockable	Randomized Control Trial	Transition from ROSC to pulseless VT/VF	24% epi vs. 12% no-epi (*p*-value = 0.03)	Fibrillations from PEA/asystole	90% epi vs. 69% no epi (*p*-value < 0.001)	Epinephrine administration was associated with higher rates of abnormal cardiac rhythm after ROSC, including ventricular tachycardia and ventricular fibrillation.

### 3.2. Dosage (Table 2)

As mentioned previously, the current recommendations in the US by the AHA is to utilize 1 mg of epinephrine per dose in cardiac arrest [5]. Studies nearly three decades ago have had conflicting data on whether this standard dosing of epinephrine or higher dosage of epinephrine is more efficacious. When comparing 0.02 mg of epinephrine per kilogram (kg) of body weight to a higher dose of 0.2 mg per kg, there was no statistically significant difference between ROSC, survival to hospital admission or discharge, or neurological outcome [14]. Similarly, comparing doses of 1 mg to 7 mg showed no statistically significant difference in cerebral performance or survival to admission or discharge, even when controlling for OHCA vs. IHCA [15]. However, a different study showed a statistically significant difference in ROSC and survival to admission between patients receiving up to 15 doses of 5 mg of epinephrine compared to standard dose of 1 mg, with no statistically significant benefit in survival to discharge or neurological outcome [16]. Similarly, 15 mg of epinephrine compared to the standard dose of epinephrine has also been shown to improve the rate of ROSC, but not survival to hospital discharge or neurological outcome [17].

These data from studies investigating different doses suggest that while higher dose epinephrine has the potential to increase rates of ROSC, it may not be meaningful in that patients may not survive to discharge or have a good neurological outcome. Based on this research, in 2000, the AHA recommended against high-dose epinephrine (0.1 mg/kg) due to the above mentioned studies showing no benefit to functional outcome, as well as the potential for harm [18].

Recent research has reinforced these findings with suggestions of harm at higher doses of epinephrine. Specifically, it was shown that there is a stepwise association with decreasing odds of survival with good neurological outcome with increasing dose of epinephrine [11]. Overall, higher doses of epinephrine have not been shown to improve meaningful survival, which has implications for prolonged resuscitations when considering neurological outcomes if patients achieve ROSC.

Several recent studies have examined the impact of cumulative doses of epinephrine in extracorporeal cardiopulmonary resuscitation (ECPR). Garcia et al. reported that, among cardiac arrest patients undergoing ECPR, those who received a low dose (less than 3 mg of epinephrine) had more favorable neurological outcomes compared to those who received a high dose (more than 3 mg) [19]. Lamhaut et al. demonstrated that, among other criteria, patients who received less than 5 mg of epinephrine during ECPR had significantly improved survival rates [20]. These findings suggest that lower cumulative doses of epinephrine, when combined with appropriate patient selection and timely ECPR initiation, may enhance survival outcomes in cardiac arrest patients.

**Table 2 medicina-60-01904-t002:** Doses of epinephrine for different outcomes among patients with cardiac arrest.

Author	Year	Country	Setting	Initial Cardiac Rhythm	Design	Primary Outcome	Primary Findings	Additional Outcomes	Additional Findings	Conclusion
Brown, et al. [14]	1992	United States	OHCA	Shockable	Randomized Control Trial	Return of Spontaneous Circulation	30% standard dose epi vs. 33% high dose epi (99% CI, −10 to 3)	Hospital Admission, Hospital Discharge, Conscious at Hospital Discharge (of survivors)	Hospital Admission: 22% standard dose vs. 22% high-dose (99% CI, −7 to 5) Hospital Discharge: 4% standard dose vs. 5% high-dose (99% CI, −4 to 2) Conscious at Hospital Discharge: 92% standard dose vs. 94% high-dose (99% CI, −20 to 16)	When comparing 0.02 mg of epinephrine per kg of body weight to a higher dose of 0.2 mg per kg, there was no statistically significant difference between ROSC, survival to hospital admission or discharge, or neurological outcome.
Stiell, et al. [15]	1992	Canada	OHCA, IHCA	Non-Shockable, Shockable	Randomized Control Trial	Survival for One Hour	23% standard dose epi vs. 18% high dose epi (95% CI, −1 to 12; *p*-value = 0.12)	ROSC, Hospital discharge, Cerebral performance	ROSC: 32% standard dose vs. 38% high dose (*p*-value = 0.15) Hospital Discharge: 5% standard dose vs. 3% high dose (95% CI, −2 to 5; *p*-value = 0.38) Cerebral Performance: 94% standard dose vs. 90% high dose (*p*-value = 0.24)	Comparing 1 mg (standard dose) of epinephrine to 7 mg (high dose) showed no statistically significant difference in ROSC, survival for one-hour or to discharge, or cerebral performance even when controlling for OHCA vs. IHCA.
Gueugniaud, et al. [16]	1998	France, Belgium	OHCA	Non-Shockable, Shockable	Randomized Control Trial	Return of Spontaneous Circulation	34.4% standard dose epi vs. 38.0% high dose epi (95% CI, 0.6 to 6.6; *p*-value = 0.02)	Hospital Admission, Hospital Discharge, Neurologic Outcomes	Hospital Admission: 23.6% standard dose vs. 26.5% high dose (*p*-value = 0.05) Hospital Discharge: 2.8% standard dose vs. 2.3% high dose (*p*-value = 0.34) Good Neurologic Outcomes: 71.7% standard dose vs. 76.3% high dose (*p*-value = 0.64)	There was a statistically significant difference in ROSC and survival to admission between patients receiving up to 15 doses of 5 mg of epinephrine compared to standard 1 mg, with no statistically significant benefit in survival to discharge or neurological outcome.
Callaham, et al. [17]	1992	United States	OHCA	Non-Shockable, Shockable	Randomized Control Trial	Return of Spontaneous Circulation	8% standard dose epi vs. 13% high dose epi (*p*-value = 0.01)	Hospital Admission, Survival to Hospital Discharge, Neurologic Outcome	Hospital Admission: 10% standard dose vs. 18% high dose (*p*-value = 0.02) Survival to Hospital Discharge: 1.2% standard dose vs. 1.7% high dose (*p*-value = 0.83) Good Neurologic Outcome: 67% standard dose vs. 0% high dose (*p*-value = 0.45)	High dose epinephrine (15 mg) compared to standard dose epinephrine was shown to improve the rate of ROSC and hospital admission, but not survival to hospital discharge or neurological outcome.
Dumas, et al. [11]	2014	France	OHCA	Non-Shockable, Shockable	Retrospective Cohort Study	Favorable Neurological Outcome—Dosage	1 mg: 31.2% 1 mg epi vs. 60.5% no epi (aOR = 0.48; 95% CI, 0.27 to 0.84; *p*-value = 0.01) 2–5 mg: 17.7% 2–5 mg vs. 60.5% no epi (aOR = 0.30; 95% CI, 0.20 to 0.47) >5 mg: 12.0% >5 mg vs. 60.5% no epi (aOR = 0.23; 95% CI, 0.14 to 0.37)	Favorable Neurological Outcome—Timing	9 min: aOR = 0.54; 95% CI, 0.32 to 0.91 10 to 15 min: aOR: 0.33; 95% CI, 0.20 to 0.56 16 to 22 min: aOR= 0.23; 95% CI: 0.12 to 0.43 >22 min: aOR: 0.17; 95% CI: 0.09 to 0.34	Administration of epinephrine was negatively associated with favorable neurological outcomes, after adjusting for confounding variables, in a stepwise fashion

### 3.3. Timing (Table 3)

The AHA recommends the use of epinephrine every 3 to 5 min in the current ACLS guidelines. However, while it appears that the early administration of epinephrine overall results in better survival and neurological outcomes, studies to determine the optimal timing of epinephrine administration have yielded varying results.

A retrospective analysis performed at 570 hospitals in the US that evaluated patients who experienced a cardiac arrest with initial rhythm of pulseless electrical activity (PEA) or asystole assessed outcomes based on time for first dose of epinephrine [21]. Earlier administration of epinephrine (as early as 1 to 3 min), has been associated with increased rates of in-hospital survival with stepwise changes for increasing intervals [21]. A similar analysis demonstrated that epinephrine administration within 10 min of EMS arrival for OHCA patients with non-shockable rhythms was associated with the highest survival, and there was a reported 4% decrease in the odds of survival to hospital discharge with each additional minute delay in administration [22]. One multicenter observational study in Japan reported similar findings among patients with ventricular fibrillation showing a significantly higher rate of neurologically intact survival if epinephrine administration again occurred within 10 min of arrest compared to no epinephrine administration [23]. There was no statistically significant difference in survival if epinephrine administration occurred greater than 10 min after arrest [23]. Two more recent propensity score-matched cohorts found similar higher rate of survival in patients receiving early epinephrine across both shockable and non-shockable rhythms [24,25].

In contrast, while early epinephrine administration (within 10 min) was associated with increased rates of ROSC in one Michigan retrospective observational study, there was no associated improvement in survival to hospital discharge [26]. The aforementioned 2018 RCT in the UK comparing the administration of epinephrine to placebo, similarly found higher rates of ROSC with epinephrine with no statistically significant difference in survival or neurological outcomes [12], suggesting on secondary analysis of timing of epinephrine specifically, that administration timing ultimately has no significant effect on these secondary outcomes [27].

A 2019 secondary analysis of the Resuscitation Outcomes Consortium’s continuous chest compression trial looked specifically at the average interval of epinephrine dosing itself and found that shorter interval times, i.e., administration less than every 3 min compared to 3 to 4 min, 4 to 5 min, and greater than 5 min, improved survival and neurological outcomes after OHCA [28]. However, the data concerning interval dosing patterns in IHCA appears to differ. Two observational analyses of adult IHCA found that more frequent epinephrine dosing, including the current guideline of administration every 3 to 5 min, led to worsened outcomes in both shockable and non-shockable rhythms [29,30]. Decreased rates of ROSC, survival to discharge, and good neurological outcomes were also seen when epinephrine was administered within the first two minutes after defibrillation in IHCA patients with a shockable rhythm, which occurred in greater than 50% of the cardiac arrests included in the cohort [31]. This observed practice is contrary to current guidelines that recommend rapid defibrillation (within 2 min) for IHCA with shockable cardiac rhythms and epinephrine administration only after multiple defibrillation attempts in refractory ventricular tachycardia or ventricular fibrillation [5,32]. Specifically, the current AHA guidelines recommend epinephrine administration after the second defibrillation in these shockable cardiac rhythms [5]. Similar cardiac arrest guidelines outlined by the 2021 European Resuscitation Council (ERC) recommend administration after the third defibrillation attempt [32]. Immediate epinephrine administration is recommended only if a non-shockable rhythm develops after the first defibrillation [5,32]. However, another study similarly found that within their patient population epinephrine was given before the first defibrillation in one in five IHCA patients (20%) with an initial shockable cardiac rhythm [33]. This epinephrine administration prior to defibrillation was also associated with decreased survival rates and worsened neurological outcomes when compared to administration after defibrillation [33]. A more recent study looking at the interval of defibrillation to epinephrine administration in OHCA however supported the earlier administration of epinephrine even in this subset of patients with shockable rhythms [34]. As the interval lengthened, i.e., increased from less than 2 min, to 2 to 4 min, then to 4 to 6 min, there was a significant decrease in outcomes such as rates of ROSC, 30-day survival, and neurological outcomes [34]. When a 4 to 6 min interval was used as a reference, the 2 to 4 min defibrillation to epinephrine interval had the highest association with better short-term outcomes, and an interval over 6 min was found to significantly lower rates of 30-day survival and good neurological outcomes [34].

Numerous potential explanations have been proposed to explain these findings, first that epinephrine administration prior to defibrillation may simply delay defibrillation itself [33]. As discussed previously, another explanation is that epinephrine administration may reduce blood flow and oxygenation to other organs by increasing oxygen demand at the myocardium, subsequently reducing ROSC and possibly lowering rates of meaningful neurological outcomes compared to defibrillation alone [6,7,33,34,35,36]. Overall, early epinephrine administration, shorter intervals between epinephrine, and defibrillation prior to epinephrine administration may improve survival, but may not lead to meaningful neurological outcomes.

**Table 3 medicina-60-01904-t003:** Timing of epinephrine for different outcomes among patients with cardiac arrest.

Author	Year	Country	Setting	Initial Cardiac Rhythm	Design	Primary Outcome	Primary Findings		Additional Outcomes	Additional Findings		Conclusion
Donnino, et al. [21]	2014	United States	IHCA	Non-Shockable	Retrospective Analysis	Survival to Hospital Discharge	12% 1–3 min to Epi (aOR = 1; 95% CI, ref) 10% 4–6 min (aOR = 0.91; 95% CI, 0.82 to 1.00; *p*-value = 0.055) 8% 7–9 min (aOR = 0.74; 95% CI, 0.63 to 0.88; *p*-value < 0.001) 7% >9 min (aOR = 0.63; 95% CI, 0.52 to 0.76; *p*-value < 0.001)		Sustained ROSC	52% 1–3 min to Epi (aOR = 1; 95% CI, ref) 47% 4–6 min (aOR = 0.90; 95% CI, 0.85 to 0.96; *p*-value < 0.001) 44% 7–9 min (aOR = 0.81; 95% CI, 0.74 to 0.89; *p*-value < 0.001) 40% >9 min (aOR = 0.70; 95% CI, 0.61 to 0.75; *p*-value < 0.001)		Earlier administration of epinephrine has been associated with increased rates of in-hospital survival with stepwise changes for increasing intervals.
									Survival to 24 h	29% 1–3 min to Epi (aOR = 1; 95% CI, ref) 26% 4–6 min (aOR = 0.91; 95% CI, 0.85 to 0.97; *p*-value = 0.005) 23% 7–9 min (aOR = 0.80; 95% CI, 0.72 to 0.89; *p*-value < 0.001) 21% >9 min (aOR = 0.70; 95% CI, 0.62 to 0.79; *p*-value < 0.001)		
									Survival with Good Neurological Outcome	7% 1–3 min to Epi (aOR = 1; 95% CI, ref) 6% 4–6 min (aOR = 0.93; 95% CI, 0.82 to 1.06; *p*-value = 0.27) 5% 7–9 min (aOR = 0.77; 95% CI, 0.62 to 0.95; *p*-value = 0.01) 4% >9 min (aOR = 0.68; 95% CI, 0.53 to 0.86; *p*-value = 0.002)		
Hansen, et al. [22]	2018	United States, Canada	OHCA	Non-Shockable	Secondary Analysis of a Prospective Study	Survival to Hospital Discharge	2.6% early (<10 min) vs. 1.7% late (≥10 min) (aOR = 0.82; 95% CI, 0.68 to 0.98)		Favorable Neurological Outcome	Reduction by 6% for each minute delay (aOR = 0.94; 95% CI, 0.89 to 0.98)		Early epinephrine administration is associated with better outcomes, including survival to hospital discharge and neurological outcome, with noticeable differences between each minute delay.
Hayashi, et al. [23]	2012	Japan	OHCA	Non-Shockable, Shockable	Observational Study	Favorable Neurological Outcome at 1 Month	Shockable Cardiac Arrest	66.7% epi ≤10 min vs. 24.9% no epinephrine (OR = 6.03; 95% CI, 1.47 to 24.69) 17.5% epi 11–20 min vs. 24.9% no epinephrine (OR = 0.64; 95% CI, 0.36 to 1.34) 6.5% epi ≥21 min vs. 24.9% no epinephrine (OR = 0.21; 95% CI, 0.09 to 0.50)	ROSC	29.3% epinephrine administration vs. 13.4% no epinephrine (*p*-value < 0.001)		There was a significantly higher rate of neurologically intact survival if epinephrine administration occurred within 10 min of shockable arrest compared to no epinephrine administration, but was not seen at longer intervals or with non-shockable rhythms.
							Non-Shockable Cardiac Arrest	0.0% epi ≤10 min vs. 3.0% no epinephrine 1.6% epi 11–20 min vs. 3.0% no epinephrine (OR = 0.52; 95% CI, 0.22 to 1.21) 1.5% epi ≥21 min vs. 3.0% no epinephrine (OR = 0.49; 95% CI, 0.21 to 1.13)				
Fukuda, et al. [24]	2021	Japan	OHCA	Non-Shockable, Shockable	Observational Study	Favorable Neurological Outcome at 1 Month	Per minute delay: OR = 0.91; 95% CI, 0.90 to 0.92; *p*-value < 0.0001		ROSC	Per minute delay: OR = 0.96; 95% CI, 0.96 to 0.96; *p*-value < 0.0001		For each minute delay in epinephrine, there was a decreased chance of neurologically favorable survival, overall survival, and prehospital ROSC.
									Overall Survival	Per minute delay: OR = 0.93; 95% CI, 0.93 to 0.94; *p*-value < 0.0001		
Okubo, et al. [25]	2021	United States, Canada	OHCA	Non-Shockable, Shockable	Cohort Study	Survival to Hospital Discharge	Shockable Cardiac Rhythm	27.7% 0–5 min to epi (RR = 1.12; 95% CI, 0.99 to 1.26) 19.3% 5–10 min (RR = 1.07; 95% CI, 0.97 to 1.17) 11.9% 10–15 min (RR = 0.80; 95% CI, 0.66 to 0.98) 7.3% 15–20 min (RR = 0.55; 95% CI, 0.33 to 0.89) 4.2% (RR = 0.13; 95% CI, 0.05 to 0.37)	Favorable Neurological Outcome	Shockable Cardiac Rhythm	22.2% 0–5 min to epi (RR = 1.07; 95% CI, 0.93 to 1.23) 15.2% 5–10 min (RR = 1.10; 95% CI, 0.99 to 1.23) 8.5% 10–15 min (RR = 0.76; 95% CI, 0.60 to 0.97) 4.9% 15–20 min (RR = 0.54; 95% CI, 0.30 to 0.99) 2.5% (RR = 0.07; 95% CI, 0.02 to 0.30)	Timing of epinephrine administration had significant effect on survival to hospital discharge, which decreased with delay in epinephrine administration for both shockable and non-shockable rhythms.
										Non-Shockable Cardiac Rhythm	1.3% 0–5 min to epi (RR = 1.26; 95% CI, 0.81 to 1.95) 1.0% 5–10 min (RR = 0.96; 95% CI, 0.74 to 1.24) 0.8% 10–15 min (RR = 0.82; 95% CI, 0.52 to 1.28) 0.5% 15–20 min (RR = 0.45; 95% CI, 0.18 to 1.11) 0.3% (RR = 0.19; 95% CI, 0.02 to 1.52)	
							Non-Shockable Cardiac Rhythm	2.8% 0–5 min to epi (RR = 1.28; 95% CI, 0.95 to 1.72) 2.4% 5–10 min (RR = 1.14; 95% CI, 0.96 to 1.34) 1.8% 10–15 min (RR = 1.01; 95% CI, 0.75 to 1.35) 1.1% 15–20 min (RR = 0.60; 95% CI, 0.31 to 1.15) 0.6% (RR = 0.36; 95% CI, 0.11 to 1.23)	ROSC	Shockable Cardiac Rhythm	60.3% 0–5 min to epi (RR = 1.16; 95% CI, 1.09 to 1.24) 50.4% 5–10 min (RR = 1.16; 95% CI, 1.11 to 1.22) 37.4% 10–15 min (RR = 1.28; 95% CI, 1.15 to 1.43) 25.7% 15–20 min (RR = 1.57; 95% CI, 1.13 to 2.19) 16.1% (RR = 1.50; 95% CI, 0.67 to 3.38)	
										Non-Shockable Cardiac Rhythm	33.8% 0–5 min to epi (RR = 1.42; 95% CI, 1.32 to 1.53) 30.7% 5–10 min (RR = 1.34; 95% CI, 1.28 to 1.39) 24.8% 10–15 min (RR = 1.42; 95% CI, 1.32 to 1.53) 18.6% 15–20 min (RR = 1.70; 95% CI, 1.42 to 2.03) 12.7% (RR = 2.14; 95% CI, 1.45 to 3.15)	
Koscik, et al. [26]	2013	United States	OHCA	Non-Shockable, Shockable	Retrospective Analysis	Return of Spontaneous Circulation (ROSC)		33% <10 min to Epi (OR = 1.78; 95% CI, 1.15 to 2.74) 23% >10 min (OR = 1.03; 95% CI, 0.95 to 1.11)	Survival to Hospital Discharge	5.3% <10 min to Epi (OR = 0.91; 95% CI, 0.35 to 2.37) 3.7% >10 min (OR = 0.92; 95% CI, 0.73 to 1.15)		Early epinephrine administration (within 10 min) was associated with increased rates of ROSC, but there was no associated improvement in survival to hospital discharge.
Perkins, et al. [12]	2018	United Kingdom	OHCA	Non-Shockable, Shockable	Randomized Controlled Trial	Rate of Survival at 30 Days	3.2% epi vs. 2.4% placebo (OR = 1.39; 95% CI, 1.06 to 1.8; *p*-value = 0.02)		Survival until Hospital Admission	23.8% epi vs. 8.0% placebo (OR = 3.59; 95% CI, 3.14 to 4.12)		Administration of epinephrine compared to placebo had higher rates of ROSC with epinephrine with no statistically significant difference in survival or neurological outcomes.
									Survival until Hospital Discharge	3.2% epi vs. 2.3% placebo (OR = 1.41; 95% CI, 1.0 to 1.86)		
									Favorable Neurologic Outcome at Discharge	2.2% epi vs. 1.9% placebo (OR = 1.18; 95% CI, 0.86 to 1.61)		
Perkins, et al. [27]	2020	United Kingdom	OHCA	Non-Shockable, Shockable	Secondary Analysis of Randomized Controlled Trial	Return of Spontaneous Circulation (ROSC)	Per minute delay, placebo: OR = 0.93; 95% CI, 0.92 to 0.95; *p* < 0.001 Per minute delay, epinephrine: OR = 0.96; 95% CI, 0.95 to 0.97; *p* < 0.001		Rate of Survival at 30 Days	Per minute delay, placebo: OR = 0.92; 95% CI, 0.89 to 0.95; *p * < 0.001 Per minute delay, epinephrine: OR = 0.90; 95% CI, 0.88 to 0.93; *p* < 0.001 Epinephrine vs. Placebo: Risk Difference = 0.009; 95% CI, 0.002 to 0.019; *p*-value = 0.103)		The effect of epinephrine on the rate of ROSC increases over time compared to placebo. However, there was no statistically significant difference in rates of survival and favorable neurological outcomes between the epinephrine and placebo groups.
									Survival to Hospital Discharge	Epinephrine vs. Placebo: Risk Difference = 0.008; 95% CI, 0.002 to 0.019; *p*-value = 0.122		
									Favorable Neurological Outcome at Survival	Epinephrine vs. Placebo: Risk Difference = 0.004; 95% CI, 0.006 to 0.013; *p*-value = 0.450		
Grunau, et al. [28]	2019	United States, Canada	OHCA	Non-Shockable, Shockable	Secondary Analysis	Favorable Neurological Outcome	Compared to <3 min (ref): 3 to <4 min: (OR = 0.40; 95% CI, 0.30 to 0.54) 4 to <5 min: (OR = 0.28; 95% CI, 0.20 to 0.38) ≥5 min (OR = 0.26; 95% CI, 0.19 to 0.35)		Survival to Admission	Compared to <3 min (ref): 3 to <4 min: (OR = 0.36; 95% CI, 0.32 to 0.41) 4 to <5 min: (OR = 0.25; 95% CI, 0.22 to 0.28) ≥5 min (OR = 0.20; 95% CI, 0.18 to 0.23)		Shorter time intervals to epinephrine administration improved survival and neurological outcomes after OHCA.
									Survival to Hospital Discharge	Compared to <3 min (ref): 3 to <4 min: (OR = 0.40; 95% CI, 0.31 to 0.50) 4 to <5 min: (OR = 0.27; 95% CI, 0.21 to 0.35) ≥5 min (OR = 0.23; 95% CI, 0.18 to 0.30)		
Wang et al. [29]	2016	Taiwan	IHCA	Non-Shockable, Shockable	Retrospective Observational Cohort Study	Survival to Hospital Discharge	OR = 0.05; 95% CI, 0.01 to 0.23; *p*-value < 0.001		Sustained ROSC	OR = 0.49; 95% CI, 0.15 to 1.58; *p*-value = 0.23		More frequent epinephrine dosing might lead to decreased survival to hospital discharge and worsened neurological function outcomes in both shockable and non-shockable rhythm IHCA.
									Survival for 24 h	OR = 0.04; 95% CI, 0.01 to 0.14; *p*-value < 0.001		
									Favorable Neurological Status	OR = 0.02; 95% CI, 0.002 to 0.16; *p*-value < 0.001		
Warren, et al. [30]	2014	United States	IHCA	Non-Shockable, Shockable	Retrospective Review of Prospectively Collected Data	Survival to Hospital Discharge	6.8% 4 to 5 min/dose (ref) 6.4% 6 to <7 min/dose (aOR = 1.41; 95% CI, 1.12 to 1.78) 5.6% 7 to <8 min/dose (aOR = 1.30; 95% CI, 1.02 to 1.65) 7.0% 8 to <9 min/dose (aOR = 1.79; 95% CI, 1.38 to 2.32) 7.4% 9 to <10 min/dose (aOR = 2.17; 95% CI, 1.62 to 2.92) *p* < 0.001 for trend		Survival to Hospital Discharge, Shockable Rhythms	7.9% 4 to <5 min/dose (aOR = 1; 95% CI, ref) 8.8% 5 to <6 min/dose (aOR = 1.34; 95% CI, 0.73 to 2.46) 11.2% 6 to <7 min/dose (aOR = 2.32; 95% CI, 1.33 to 4.05) 8.5%% 7 to <8 min/dose (aOR = 2.27; 95% CI, 1.18 to 4.37) 10.2% 8 to <9 min/dose (aOR = 2.66; 95% CI, 1.36 to 5.21) 12.9% 9 to <10 min/dose (aOR = 4.00; 95% CI, 1.88 to 8.52) *p* < 0.001 for trend		Higher average dosing intervals (i.e., less frequent dosing) of epinephrine than current ACLS guidelines was associated with higher rates of survival to discharge across shockable and non-shockable cardiac rhythms.
									Survival to Hospital Discharge, Non-Shockable Rhythms	6.7% 4 to <5 min/dose (aOR = 1; 95% CI, ref) 5.7% 6 to <7 min/dose (aOR = 1.34; 95% CI, 1.04 to 1.73) 5.1%% 7 to <8 min/dose (aOR = 1.17; 95% CI, 0.90 to 1.52) 6.5% 8 to <9 min/dose (aOR = 1.64; 95% CI, 1.20 to 2.22) 6.4% 9 to <10 min/dose (aOR = 1.97; 95% CI, 1.43 to 2.71) *p* < 0.001 for trend		
Andersen, et al. [31]	2016	United States	IHCA	Shockable	Prospective Observational Cohort Study	Survival to Hospital Discharge	31% early epi vs. 48% (aOR = 0.70; 95% CI, 0.59 to 0.82; *p*-value < 0.001)		ROSC	67% early epi vs. 79% (aOR = 0.71; 95% CI, 0.60 to 0.83; *p*-value < 0.001)		There were decreased odds of survival to hospital discharge, ROSC, and neurological outcomes at discharge when epinephrine was administered within two minutes of the first defibrillation.
									Good Functional Outcome	25% early epi vs. 41% (aOR = 0.69, 95% CI, 0.58 to 0.83; *p*-value < 0.001)		
Evans, et al. [33]	2021	United States	IHCA	Shockable	Propensity matched analysis	Survival to Hospital Discharge	22.4% epi before defibrillation vs. 29.7% (OR = 0.69; 95% CI, 0.64 to 0.74; *p*-value < 0.001)		Favorable Neurological Survival	15.8% vs. 21.6% (OR = 0.68; 95% CI, 0.61 to 0.76; *p*-value < 0.001)		One in five patients with IHCA are treated with epinephrine prior to initial defibrillation contrary to current recommendations, and this practice is associated with decreased survival to discharge.
									Survival after Acute Resuscitation	61.7% vs. 69.5% (OR = 0.73; 95% CI 0.67 to 0.79; *p*-value < 0.001)		
Kawakami, et al. [34]	2024	Japan	OHCA	Shockable	Retrospective Analysis	Favorable Neurological Outcome	15.4% 4- < 6 min defibrillation to epi (aOR = 1, 95% CI, ref) 10.4% 6- < 8 min (aOR = 0.75, 95% CI, 0.67 to 0.85) 9.7% 8- < 10 min (aOR = 0.61, 95% CI, 0.54 to 0.69) 6.7% 10- < 12 min (aOR = 0.49, 95% CI, 0.43 to 0.56) 6.4% 12- < 14 min (aOR = 0.42, 95% CI, 0.37 to 0.49) 4.8% 14- < 16 min (aOR = 0.33, 95% CI, 0.29 to 0.39) 4.3% 16- < 18 min (aOR = 0.29, 95% CI, 0.25 to 0.35) 3.2% ≥18 min (aOR = 0.23, 95% CI, 0.20 to 0.27) *p* < 0.001 for trend		ROSC	36.9% 4- < 6 min (aOR = 1, 95% CI, ref) 30.9% 6- < 8 min (aOR = 0.73, 95% CI, 0.63 to 0.83) 26.5% 8- < 10 min (aOR = 0.68, 95% CI, 0.59 to 0.78) 22.7% 10- < 12 min (aOR = 0.53, 95% CI, 0.46 to 0.61) 19.9% 12- < 14 min (aOR = 0.53, 95% CI, 0.46 to 0.62) 16.3% 14- < 16 min (aOR = 0.40, 95% CI, 0.34 to 0.48) 14.7% 16- < 18 min (aOR = 0.32, 95% CI, 0.27 to 0.39) 11.8% ≥18 min (aOR = 0.24, 95% CI, 0.20 to 0.28) *p* < 0.001 for trend		A longer defibrillation to epinephrine level is associated with a significant decrease in outcomes such as rates of ROSC, 30-day survival, and neurological outcomes, supporting earlier administration of epinephrine in OHCA patients with shockable rhythms.
									Survival at 30 Days	28.6% 4- < 6 min (aOR = 1, 95% CI, ref) 22.8% 6- < 8 min (aOR = 0.63, 95% CI, 0.53 to 0.76) 21.1% 8- < 10 min (aOR = 0.61, 95% CI, 0.51 to 0.73) 17.1% 10- < 12 min (aOR = 0.42, 95% CI, 0.34 to 0.51) 16.9% 12- < 14 min (aOR = 0.41, 95% CI, 0.33 to 0.50) 13.2% 14- < 16 min (aOR = 0.30, 95% CI, 0.24 to 0.38) 11.2% 16- < 18 min (aOR = 0.26, 95% CI, 0.20 to 0.35) 7.9% ≥18 min (aOR = 0.21, 95% CI, 0.16 to 0.26) *p* < 0.001 for trend		

### 3.4. Route of Administration (Table 4)

Lack of vascular access in a cardiac arrest can delay the administration of epinephrine. While interosseous (IO) access is a good option in patients whose intravenous (IV) access is not feasible or when prior attempts were unsuccessful, the procedure still takes time to establish access. However, studies have shown that IO administration of epinephrine has no difference in effect, including ROSC, survival, or neurological outcome, compared to IV administration in OHCA [37,38].

An alternative route is intramuscular (IM) epinephrine administration. A recent single-center study with a before-and-after study design utilized IM epinephrine in addition to typical protocols for IV epinephrine [39]. This study used 5 mg of IM epinephrine, which was calculated to be equivalent to 0.5 mg of IV epinephrine [39]. The results showed that an initial IM dose of epinephrine was associated with increased rates of survival to admission, survival to hospital discharge, and neurological outcome [39].

Another route recommended by the AHA in children experiencing cardiac arrest without IV or IO access is endotracheal administration prior to establish vascular access [5]. When used prior to IV access being obtained and thus prior to IV epinephrine, ET epinephrine in infants has been shown to have higher rates of ROSC compared to IV alone or ET administration alone [40]. However, endotracheal administration does not appear to have survival benefits in adults [41].

**Table 4 medicina-60-01904-t004:** Route of administration of epinephrine for different outcomes among patients with cardiac arrest.

Author	Year	Country	Setting	Initial Cardiac Rhythm	Design	Primary Outcome	Primary Findings	Additional Outcomes	Additional Findings	Conclusion
Tan, et al. [37]	2021	Singapore	OHCA	Non-Shockable, Shockable	Prospective; Parallel-Group, Cluster-Randomized Study	Return of Spontaneous Circulation Pre-Hospital	11.7% IV/IO vs. 11.7% IV only (aOR = 0.99; 95% CI, 0.75 to 1.29; *p*-value = 0.998)	Vascular access rates, Time to administration, Survival Outcomes	Vascular Access Rates: 76.6% IV/IO vs. 61.1% IV only (*p*-value = 0.001) Time to Epi: 23 min IV/IO vs. 25 min IV (*p*-value = 0.001) Survival to Hospital Discharge: 4.9% IV/IO vs. 8.4% IV only (*p*-value = 0.027) Good Neuro Outcome: 3.4% IV/IO vs. 4% IV only (*p*-value = 0.630)	Although a fast option for vascular access, IO administration of epinephrine has no benefit to IV alone in OHCA, including ROSC, survival, or neurological outcome.
Nolan, et al. [38]	2020	United Kingdom	OHCA	Non-Shockable, Shockable	Secondary Analysis of Randomized Control Trial	Return of Spontaneous Circulation	IV: 26% IV epi vs. 8.6% IV placebo (OR = 4.07; 95% CI, 3.42 to 4.85) IO: 16% IO epi vs. 4.9% IO placebo (OR = 3.98; 95% CI, 2.86 to 5.53) Comparison: Interaction OR 0.98; 95% CI, 0.67 to 1.42; *p*-value = 0.90	Survival at 30 Days, Favorable Neurological Outcomes	Survival at 30 Days, IV: 4.2% IV epi vs. 2.7%IV placebo (OR = 1.67; 95% CI, 1.18 to 2.35) Survival at 30 days, IO: 1.1% IO epi vs. 1.2% IO placebo (OR = 0.90; 95% CI, 0.4 to 2.05) Survival at 30 Days, Comparison: Interaction OR 0.54; 95% CI, 0.22 to 1.32, *p*-value = 0.18 Neurological Outcomes, IV: 2.8% IV epi vs. 2.1% IV placebo (OR = 1.39; 95% CI, 0.93 to 2.06) Neurological Outcomes, IO: 0.7%% IO epi vs. 1.0%% IO placebo (OR = 0.62; 95% CI, 0.23 to 1.67) Neurological Outcomes, Comparison: Interaction OR 0.45; 95% CI, 0.15 to 1.30; *p*-value = 0.14	There was no significant difference between IO and IV administration on ROSC, 30-day survival, or favorable neurological outcomes.
Palatinus, et al. [39]	2024	United States	OHCA	Non-Shockable, Shockable	Before-and-After Implementation Study	Survival to Hospital Admission	37.1% IM vs. 31.6% standard (aOR = 1.37; 95% CI, 1.06 to 1.77)	Hospital Survival, Favorable Neurological Status at Hospital Discharge	Hospital Survival: 11.0% IM vs. 7.0% standard (aOR = 1.73; 95% CI, 1.10 to 2.71) Neurological Status: 9.8% IM vs. 6.2% standard (aOR = 1.72; 95% CI, 1.07 to 2.76)	An initial intramuscular dose of epinephrine was associated with increased rates of survival to admission, survival to hospital discharge, and neurological outcome.
Niemann and Stratton [41]	2000	United States	OHCA	Non-Shockable	Retrospective Review	Return of Spontaneous Circulation	16.7% IV vs. 0% ET (*p*-value = 0.005)	Survival to Hospital Discharge	2.6% IV vs. 0% ET	Endotracheal administration of epinephrine does not appear to have survival benefits in adults compared to intravenous administration.

Overall, IM and IO administration of epinephrine have shown to be effective in improving survival similarly to IV use. However, additional studies need to be performed to establish an appropriate route of administration, depending on available vascular access.

### 3.5. Epinephrine and Other Vasopressors (Table 5)

Utilizing other vasopressors during cardiac arrest has also been studied to determine if they can improve outcomes. A small RCT based in Canada and London utilized either one intravenous dose of vasopressin (40 U) or epinephrine (1 mg) when the first dose of epinephrine would be given within the ACLS algorithm, with subsequent doses of epinephrine used if the patient did not achieve ROSC [42]. There was no statistically significant difference between initial vasopressin or epinephrine use in survival to hospital discharge or in neurological outcome [42]. Another RCT however did find that patients with asystole who received vasopressin over epinephrine had higher rates of hospital admission and hospital discharge, but did not have any statistically significant difference in neurological outcomes [43]. A larger multicenter study evaluated vasopressin versus epinephrine and found that there was no statistically significant difference in survival to hospital discharge or neurological outcomes [44]. However, when looking at a sub-group analysis, those with PEA rhythm showed higher survival to admission for the vasopressin group [44]. When using vasopressin combined with epinephrine in another study, there was no survival benefit or change in neurological outcome [45]. A recent meta-analysis determined that while epinephrine showed superiority over other vasopressors in survival to hospital discharge and ROSC, there was no benefit in neurological outcome [46]. Overall, the use of vasopressin in cardiac arrest may offer survival benefits over epinephrine, but with questions about functional neurological status.

**Table 5 medicina-60-01904-t005:** Epinephrine versus other vasopressors on outcomes among patients with cardiac arrest.

Author	Year	Country	Setting	Initial Cardiac Rhythm	Design	Primary Outcome	Primary Findings	Additional Outcomes	Additional Findings	Conclusion
Stiell, et al. [42]	2001	Canada	IHCA	Non-Shockable, Shockable	Triple-Blind Randomized Controlled Trial	1 Hour Survival	39% vasopressin vs. 35% epi (95% CI, −10.9% to 17.0%; *p*-value = 0.66)	Survival to Hospital Discharge; Neurological Function	Survival to Hospital Discharge: 12% vasopressin vs. 14% epinephrine (95% CI, −11.8% to 7.8%; *p*-value = 0.67) Neurological Function (MMSE): 36 vaso vs. 35 epi (*p*-value = 0.75)	There was no statistically significant difference between initial vasopressin or epinephrine use in survival to hospital discharge or in neurological outcome.
Wenzel, et al. [43]	2004	Austria, Germany, Switzerland	OHCA	Non-Shockable, Shockable	Double-Blind; prospective, Multicenter, Randomized Controlled Trial	Survival to Hospital Admission, Asystole	29.0% vasopressin vs. 20.3% epinephrine (*p*-value = 0.02)	Survival to Hospital Discharge, Asystole; Survival to Hospital Admission, Ventricular Fibrillation; Survival to Hospital Admission, PEA; Cerebral Performance	Survival to Hospital Discharge, Asystole: 4.7% vaso vs. 1.5% epi (*p*-value = 0.04) Survival to Hospital Admission, Ventricular Fibrillation: 46.2% vaso vs. 43.0% epi (*p*-value = 0.48) Survival to Hospital Admission, PEA: 33.7% vaso vs. 30.5% epi (*p*-value = 0.65) Cerebral Performance: 32.6% vaso vs. 34.8% epi (*p*-value = 0.99)	Patients with asystole who received vasopressin over epinephrine had higher rates of hospital admission and hospital discharge, but overall there were no statistically significant difference in neurological outcomes.
Ong, et al. [44]	2012	Singapore	OHCA	Non-Shockable, Shockable	Double-Blind, Multicenter; parallel-Design, Randomized Controlled Trial	Survival to Hospital Discharge	2.3% epi vs. 2.9% vasopressin (RR = 1.72; 95% CI, 0.65 to 4.51; *p*-value = 0.27)	Return of Spontaneous Circulation, Survival to Admission, Neurological Status	ROSC: 30.0% epi vs. 31.8% vasopressin (RR = 1.15; 95% CI, 0.87 to 1.52, *p* = 0.33) Admission: 22.2% vasopressin vs. 16.7% epi (RR = 1.18, 95% CI, 1.00 to 1.38, *p*-value = 0.06) Neuro Status: 2.3% epi vs. 2.9% vasopressin	There was no statistically significant difference in survival to hospital discharge or neurological outcomes with use of epinephrine versus vasopressin. However, when looking at sub-group analysis, there appeared to be higher survival to admission for the vasopressin group.
Kim, et al. [45]	2022	Republic of Korea	OHCA	Non-Shockable, Shockable	Double-Blind, Single-Center, Randomized; Placebo-Controlled Trial	Return of Spontaneous Circulation	36.5% vasopressin + epi vs. 32.4% placebo + epi (RR = 0.94; 95% CI, 0.74 to 1.19; *p*-value = 0.60)	Survival to Discharge, Neurologic Status	Survival to Discharge: 8.1% vasopressin + epi vs. 8.1% placebo + epi (RR = 1.00; 95% CI, 0.91 to 1.10; *p*-value = 1.00) Neuro Status: 0.0% vasopressin + epi vs. 0.0% placebo + epi (RR = 1.00; 95% CI, 1.00 to 1.00; *p*-value = 1.00)	When using vasopressin combined with epinephrine there was no survival benefit or change in neurological outcome.
Chandler, et al. [46]	2024	Various	OHCA, IHCA	Non-Shockable, Shockable	Meta-Analysis	Survival to Hospital Discharge	OR = 1.52; 95% CI, 1.20 to 1.94; *p* value < 0.001	Return of Spontaneous Circulation, Survival to Hospital Admission, Survival to Hospital Discharge, Neurological Outcomes, Myocardial Infarction, and Incidence of Arrhythmias	Survival at 30 Days: OR = 1.58; 95% CI, 1.42 to 1.76; *p*-value < 0.00001 ROSC: OR = 3.60; 95% CI, 3.45 to 3.76; *p*-value < 0.00001 Neurological Function: OR = 1.31; 95% CI, 0.99 to 1.73; *p*-value = 0.06	While epinephrine showed superiority over other vasopressors in survival to hospital discharge and ROSC, there was no benefit in neurological outcome.

## 4. Discussion

The efficacy of epinephrine use in cardiac arrest has been studied and debated, despite the medication being a cornerstone of treatment in standardized cardiac arrest protocols. Specifically, it is questioned whether the use of epinephrine actually provides patients with meaningful neurological recovery following initial resuscitation. Several trials suggest that epinephrine may assist with ROSC itself but is unlikely to help patients survive to hospital discharge, particularly with good neurological outcomes [8,9,10,11]. Furthermore, higher doses of epinephrine above 1 mg per administration are unlikely to offer survival benefit and may actually cause harm [14,15,16,17]. Earlier administration of epinephrine in OHCA has been found in many cases to have improved rates of ROSC, survival to discharge, and neurological outcomes; however, compared to placebo, epinephrine did not account for a significant difference in these secondary outcomes regardless of administration timing or interval [12,27]. While the current recommendations by the AHA suggest the administration of epinephrine every 3 to 5 min [5], shorter intervals may improve outcomes in cases of OHCA but worsen outcomes in IHCA. Regarding the route of administration, IM use may be a good option for medication administration when IV or IO access is unable to be obtained although additional data are needed [39].

While epinephrine is currently regarded as the standard of care in cardiac resuscitation, future studies should include more randomized control trials on the efficacy of epinephrine use in cardiac arrest compared to not using epinephrine (i.e., placebo versus alternate vasopressors) to determine the impact on meaningful survival. More comparative data across practice settings and initial cardiac rhythms is needed to ensure chosen pharmacological interventions are not only able to truly improve patient outcomes but are not detrimental to the patient’s condition.

This area of research has the potential to significantly impact resuscitation protocols in cardiac arrest. Cardiac arrest is a multifaceted condition of which the exact underlying cause is not often known in the acute setting. Patients are commonly stratified for the cardiac arrest algorithm based on initial cardiac rhythm, which also does not provide in-depth information about the underlying pathology. Despite often having a paucity of additional patient history, a limited physical exam, and little to no laboratory results when initiating the ACLS algorithm, especially in OHCA, the current protocol still relies on epinephrine almost exclusively as the pharmacological arm to treat a potential combination of complex disease processes. It is perhaps unreasonable to assume one medication can have uniform efficacy across pathologies, which is seen in the variable data on epinephrine administration across practice settings and shockable versus non-shockable rhythms. In addition to RCTs on epinephrine use compared to placebo in the current algorithm, future studies should also look to revolutionize our protocols by expanding knowledge about critical conditions leading to cardiac arrest and how these can particularly be targeted during a resuscitation. One way that may help guide resuscitations is through the use of Artificial Intelligence (AI), and future projects should investigate how the diagnostic capabilities of AI may provide key information to help target patient care in cardiac arrest.

## 5. Limitations

This study is a focused narrative literature review on the efficacy, timing, and dosage of epinephrine use in cardiac arrest. The intention was not to be a systematic review. A primary limitation to data available for review is that there is a limited number of RCTs evaluating the use of epinephrine in the management of cardiac arrest. This is likely because cardiac arrest has high morbidity and mortality and the ethics of withholding possibly life-saving medication by not using epinephrine are complicated. Additionally, these studies primarily looked at OHCA, with more limited data on IHCA where there are more resources and a faster onset of good quality CPR and epinephrine administration, a concern brought up by others regarding one of the RCTs [47]. Furthermore, while these studies looked for statistically significant differences in the use of epinephrine, they rarely include information about the number needed to treat (NNT) for a survival benefit of using epinephrine in cardiac arrest. However, for those that did mention NNT, as Perkins et al. noted in their study in 2018, the number of patients needed to treat with epinephrine to prevent one death at 30 days from cardiac arrest was 112, which is larger than previously reported numbers for bystander CPR, early defibrillation, and early recognition of cardiac arrest [12]. It is important to note that while an intervention may not have statistical significance, there may still be clinical significance, particularly in high morbidity and mortality conditions like cardiac arrest. Inherent bias may also exist as data were reviewed by different reviewers for inclusion in this study.

## 6. Conclusions

Epinephrine is still considered to be the cornerstone of pharmacologic management in cardiac arrest. However, studies have conflicting data on the efficacy of its use, and particularly on the likelihood of good neurological outcomes. Ultimately, several studies suggest that epinephrine use may not provide patients who survive cardiac arrest with a meaningful neurological recovery. Given the significant morbidity and mortality of cardiac arrest, there is a significant need for additional RCTs in different practice settings to further elucidate the efficacy of epinephrine and determine appropriate dosing and timing of epinephrine use in cardiac arrest.

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
