# Peer review of "Use of Epinephrine in Cardiac Arrest: Advances and Future Challenges"

_medicina, 2024, doi:10.3390/medicina60111904_

Round 1
Reviewer 1 Report
Comments and Suggestions for Authors
I have reviewed the manuscript entitled ‘Use of Epinephrine in Cardiac Arrest: Advances and Future Challenges’.
The role of epinephrine in cardiac arrest has been investigated well and the study is completely well-designed. The question how should we use or not use in order to increase the positive outcomes.
Please emphasize the main deductions in the study in a clearer way.
The role of inotropic drugs especially epinephrine should be emphasized in a good manner. Since the mortality rates are very high in patients with cardiogenic shock. Please consider citing ‘Evaluation of Intermountain Risk Score for Short- and Long-Term Mortality in ST Elevation Myocardial Infarction Patients’ and ‘Prognostic value of Intermountain Risk Score for short- and long-term mortality in patients with cardiogenic shock’.
Please explain the inclusion and exclusion criteria more in detail.
The role of AI intelligence in these cardiac arrest patients should be discussed in a short section citing ‘The Role of Artificial Intelligence in Coronary Artery Disease and Atrial Fibrillation’
Author Response
Reviewer 1:
I have reviewed the manuscript entitled ‘Use of Epinephrine in Cardiac Arrest: Advances and Future Challenges’.
The role of epinephrine in cardiac arrest has been investigated well and the study is completely well-designed. The question how should we use or not use in order to increase the positive outcomes.
Please emphasize the main deductions in the study in a clearer way.
The role of inotropic drugs especially epinephrine should be emphasized in a good manner. Since the mortality rates are very high in patients with cardiogenic shock. Please consider citing ‘Evaluation of Intermountain Risk Score for Short- and Long-Term Mortality in ST Elevation Myocardial Infarction Patients’ and ‘Prognostic value of Intermountain Risk Score for short- and long-term mortality in patients with cardiogenic shock’.
Please explain the inclusion and exclusion criteria more in detail.
The role of AI intelligence in these cardiac arrest patients should be discussed in a short section citing ‘The Role of Artificial Intelligence in Coronary Artery Disease and Atrial Fibrillation’
Authors’ Response to Reviewer 1:
Thank you for taking the time to review our review and for providing suggestions for improvement. In order to emphasize the main deductions, we have made sure to summarize each section with some additions to the end of each subsection. We have also expanded the inclusion and exclusion criteria.
Our new inclusion and exclusion criteria are now: “Our review focused on studies that met stringent inclusion criteria including adult cardiac arrest patients who received epinephrine during their resuscitation. We included randomized controlled trials (RCTs), prospective observational studies, and secondary analyses of prospective observational data . We also included retrospective studies to strengthen our arguments regarding supplementary interventions in cardiac arrest management, such as extracorporeal cardiopulmonary resuscitation (ECPR). The primary criterion was that these studies evaluated therapeutic interventions involving the administration of epinephrine during cardiac arrest and reported patient-related outcomes. These outcomes could include survival rates, neurological function, return of spontaneous circulation (ROSC), and other relevant clinical endpoints that assessed the effectiveness and safety of epinephrine in cardiac arrest….. Studies prior to 1990 were also excluded. Furthermore, as this review focused on cardiac arrest in adult patients, studies with participants only under the age of 18 were excluded.”
.
Thank you for your comments on the role of cardiogenic shock and AI, however, the focus of this paper is not on CAD nor Atrial Fibrillation, nor cardiogenic shock, and thus we did not talk about the role of AI in those disease processes. Furthermore, the suggestions to include risk scores for STEMI and cardiogenic shock do not pertain directly to cardiac arrest or the use of epinephrine in cardiac arrest, so is beyond the scope of this review. Upon careful review of the suggested articles, we apologize that we cannot include them in our revision as those articles were not related to the topic of this manuscript.
Reviewer 2 Report
Comments and Suggestions for Authors
This study is a focused narrative literature review on efficacy, timing, and dosage of epinephrine use in cardiac arrest. It is not a systematic review. The reason is that there is a limited number of RCTs evaluating the use of epinephrine in management of cardiac arrest. The results showed that earlier administration of epinephrine in cardiac arrest is more likely to have improved outcomes compared to later administration and longer intervals. Intravenous is the preferred route of administration for epinephrine, but new research suggests intramuscular administration may be beneficial. While epinephrine has been shown to improve rates of return of spontaneous circulation and even survival to hospital discharge, epinephrine use may not provide patients who survive cardiac arrest with a meaningful neurological recovery. The authors made one observation about the number needed to treat (NNT) for a survival benefit of using epinephrine in cardiac arrest which was 112, which is larger than previously reported numbers for bystander CPR. Lastly the authors mentioned that while something may not have statistical significance, there may still be clinical significance, particularly in high morbidity and mortality conditions like cardiac arrest.
The methodology of this paper is GREAT. The data presentation is clear and effectively enhances understanding of the problem
Author Response
Reviewer 2:
This study is a focused narrative literature review on efficacy, timing, and dosage of epinephrine use in cardiac arrest. It is not a systematic review. The reason is that there is a limited number of RCTs evaluating the use of epinephrine in management of cardiac arrest. The results showed that earlier administration of epinephrine in cardiac arrest is more likely to have improved outcomes compared to later administration and longer intervals. Intravenous is the preferred route of administration for epinephrine, but new research suggests intramuscular administration may be beneficial. While epinephrine has been shown to improve rates of return of spontaneous circulation and even survival to hospital discharge, epinephrine use may not provide patients who survive cardiac arrest with a meaningful neurological recovery. The authors made one observation about the number needed to treat (NNT) for a survival benefit of using epinephrine in cardiac arrest which was 112, which is larger than previously reported numbers for bystander CPR. Lastly the authors mentioned that while something may not have statistical significance, there may still be clinical significance, particularly in high morbidity and mortality conditions like cardiac arrest.
The methodology of this paper is GREAT. The data presentation is clear and effectively enhances understanding of the problem
Authors’ Response to Reviewer 2:
Thank you very much for reviewing our discussion of cardiac arrest and for the positive feedback. We have added more context and tables to make the manuscript stronger.
Reviewer 3 Report
Comments and Suggestions for Authors
Thank you for inviting me to review this work.
I find the review very interesting, as it touches on a topic of personal interest and one that sparks controversy in almost all areas of practice, continually fueling debate. The document aligns well with existing clinical trials where adrenaline doses have been limited in patients undergoing extracorporeal cardiopulmonary resuscitation (ECPR). For instance, in the study "ECPR for Refractory Out-of-Hospital Cardiac Arrest" (CHEER Trial), a protocol was established to cap adrenaline administration at a maximum of 3 mg during resuscitation before ECPR cannulation. This approach aimed to minimize potential adverse effects associated with high doses of adrenaline in the ECPR context.
In Lamhaut’s study (A Pre-Hospital Extracorporeal Cardio Pulmonary Resuscitation (ECPR) strategy for treatment of refractory out-of-hospital cardiac arrest: An observational study and propensity analysis), the total dose of adrenaline was similarly limited to 3 mg in patients undergoing ECPR. This strategy aimed to reduce potential adverse effects associated with higher doses of adrenaline in extracorporeal resuscitation. Importantly, the study observed improved clinical outcomes in the group with limited adrenaline dosing, suggesting that a lower dose of adrenaline may correlate with better outcomes in ECPR settings, possibly by reducing the side effects of high-dose adrenaline when combined with extracorporeal support.
Additionally, in the "Advanced Reperfusion Strategies for Refractory Cardiac Arrest" (ARREST Trial), although no strict limitation on adrenaline dosing was specified, the importance of high-quality resuscitation was emphasized. The trial also considered the possibility of reducing adrenaline doses for patients receiving ECPR, based on the hypothesis that extracorporeal circulation could compensate for the need for high doses of vasopressors.
These studies reflect a current trend in research toward re-evaluating optimal adrenaline doses in the ECPR context, aiming to improve clinical outcomes and reduce potential side effects.
I encourage the authors to consider this suggestion and expand the document, as I believe it would enrich the scientific content of the present review.
Author Response
Reviewer 3:
I find the review very interesting, as it touches on a topic of personal interest and one that sparks controversy in almost all areas of practice, continually fueling debate. The document aligns well with existing clinical trials where adrenaline doses have been limited in patients undergoing extracorporeal cardiopulmonary resuscitation (ECPR). For instance, in the study "ECPR for Refractory Out-of-Hospital Cardiac Arrest" (CHEER Trial), a protocol was established to cap adrenaline administration at a maximum of 3 mg during resuscitation before ECPR cannulation. This approach aimed to minimize potential adverse effects associated with high doses of adrenaline in the ECPR context.
In Lamhaut’s study (A Pre-Hospital Extracorporeal Cardio Pulmonary Resuscitation (ECPR) strategy for treatment of refractory out-of-hospital cardiac arrest: An observational study and propensity analysis), the total dose of adrenaline was similarly limited to 3 mg in patients undergoing ECPR. This strategy aimed to reduce potential adverse effects associated with higher doses of adrenaline in extracorporeal resuscitation. Importantly, the study observed improved clinical outcomes in the group with limited adrenaline dosing, suggesting that a lower dose of adrenaline may correlate with better outcomes in ECPR settings, possibly by reducing the side effects of high-dose adrenaline when combined with extracorporeal support.
Additionally, in the "Advanced Reperfusion Strategies for Refractory Cardiac Arrest" (ARREST Trial), although no strict limitation on adrenaline dosing was specified, the importance of high-quality resuscitation was emphasized. The trial also considered the possibility of reducing adrenaline doses for patients receiving ECPR, based on the hypothesis that extracorporeal circulation could compensate for the need for high doses of vasopressors.
These studies reflect a current trend in research toward re-evaluating optimal adrenaline doses in the ECPR context, aiming to improve clinical outcomes and reduce potential side effects.
I encourage the authors to consider this suggestion and expand the document, as I believe it would enrich the scientific content of the present review.
Authors’ Response to Reviewer 3:
Thank you for your great suggestions, we added a paragraph citing these studies and discussing the lower cumulative doses of epinephrine, when combined with appropriate patient selection and timely ECPR initiation, may enhance survival outcomes in cardiac arrest patients.
Our new paragraph was added to the Section about dosage.
The new paragraph now reads: “Recent research has reinforced these findings with suggestions of harm at higher doses of epinephrine. Specifically, it was shown that there is a stepwise association with decreasing odds of survival with good neurological outcome with increasing dose of epi-nephrine.11 Overall, higher doses of epinephrine have not been shown to improve meaningful survival, which has implications for prolonged resuscitations when considering neurological outcomes if patients achieve ROSC.
Several recent studies have examined the impact of cumulative doses of epinephrine in extracorporeal cardiopulmonary resuscitation (ECPR). Garcia et al. reported that, among cardiac arrest patients undergoing ECPR, those who received a low dose (less than 3 mg of epinephrine) had more favorable neurological outcomes compared to those who received a high dose (more than 3 mg).19 Lamhaut et al. demonstrated that, among other criteria, patients who received less than 5 mg of epinephrine during ECPR had significantly improved survival rates.20 These findings suggest that lower cumulative doses of epinephrine, when combined with appropriate patient selection and timely ECPR initiation, may enhance survival outcomes in cardiac arrest patients.”